# Constant Force-Tracking Control Based on Deep Reinforcement Learning in Dynamic Auscultation Environment

**DOI:** 10.3390/s23042186

**Published:** 2023-02-15

**Authors:** Tieyi Zhang, Chao Chen, Minglei Shu, Ruotong Wang, Chong Di, Gang Li

**Affiliations:** 1School of Mathematics and Statistics, Qilu University of Technology (Shandong Academy of Sciences), Jinan 250353, China; 2Shandong Artificial Intelligence Institute, Qilu University of Technology (Shandong Academy of Sciences), Jinan 250014, China

**Keywords:** constant force-tracking, deep reinforcement learning, auscultation robot, compliant control

## Abstract

Intelligent medical robots can effectively help doctors carry out a series of medical diagnoses and auxiliary treatments and alleviate the current shortage of social personnel. Therefore, this paper investigates how to use deep reinforcement learning to solve dynamic medical auscultation tasks. We propose a constant force-tracking control method for dynamic environments and a modeling method that satisfies physical characteristics to simulate the dynamic breathing process and design an optimal reward function for the task of achieving efficient learning of the control strategy. We have carried out a large number of simulation experiments, and the error between the tracking of normal force and expected force is basically within ±0.5 N. The control strategy is tested in a real environment. The preliminary results show that the control strategy performs well in the constant force-tracking of medical auscultation tasks. The contact force is always within a safe and stable range, and the average contact force is about 5.2 N.

## 1. Introduction

Medical auscultation [1] is an important part of respiratory system examination. Through the analysis of breath sounds, it is possible to diagnose various respiratory system diseases. In the new medical mode, in order to collect auscultation information, the whole auscultation process is completed via a constant force-tracking control manipulator. At the same time, sound collection is realized through a stethoscope fixed at the end of the manipulator. In this process, the manipulator needs to apply the appropriate force to ensure a clear collection of individual sounds to the chest, abdomen or back. However, due to the fluctuation of the body caused by breathing, the position of the auscultation site changes, and the skin also undergoes elastic deformation. Only the appropriate force and position can guarantee clear auscultation quality and a more comfortable auscultation experience [2]. Therefore, it is extremely important to control the manipulator for constant force-tracking in a dynamic environment.

In recent years, the application of deep reinforcement learning [3,4,5,6,7,8] in the robot field [9,10] has deepened and has been widely used in grasping [11,12], assembly [13], path planning [14,15], and other fields [16,17]. A few scholars have used deep reinforcement learning to study the constant force-tracking process, showing the great potential for applying deep reinforcement learning to solving the issue of constant force-tracking. However, there is still a lack of application in a dynamic environment. This paper expands the research in this area.

In this paper, we propose a constant force-tracking framework based on deep reinforcement learning based on the application scenario of medical auscultation. At the same time, in order to better achieve the purpose of a training strategy, a dynamic simulation-modeling method is designed to simulate the fluctuation of auscultation position caused by breathing. The trained model performs well during constant force-tracking, and the error between the normal tracking force and the expected force is maintained within the range of ±0.5 N. Considering the differences in body and respiratory rates between different patients, different situations can be simulated by changing the stiffness [18] and frequency settings of the model, thereby increasing the authenticity of the simulation. In order to verify the applicability of the proposed method to the constant force-tracking task, it is compared with the two methods, and a large number of generalization experiments are carried out to prove that it achieves definite generalization.

The rest of this paper is organized as follows. In Section 2, the related work in the field of constant force-tracking is introduced. In Section 3, the related principles of deep reinforcement learning, dynamic modeling methods, and details of the constant force-tracking strategy based on approximate policy optimization (ppo) [19] algorithms are described. In Section 4, the experimental setup and simulation results are described. In Section 5, a real experiment is carried out. Finally, Section 6 summarizes this paper.

## 2. Related Work

In interactive tasks such as machine auscultation [20], in order to realize the constant force-tracking of the manipulator, the manipulator is required not only to track the planned motion trajectory [21], but also to control the force interacting with the environment. This leads to complex force-control requirements, and it is critical to maintain a constant contact force during trajectory-tracking. In this section, we will review some related work on constant force-tracking in the field of impedance control and deep reinforcement learning.

### 2.1. Impedance Control Method

Conventional impedance control [22] is a force- and position-dependent dynamic control algorithm which can be used for constant force-tracking. It is a kind of compliance control method [23]. This so-called compliance refers to the ability of a robot to adapt to external changes. When a robot contacts the environment, even if the environment changes, i.e., if the contact position changes, the robot still maintains a predetermined contact force with the environment, which is its adaptability. Impedance control achieves the stabilization of the contact force by controlling the relationship between force and contact position.

On the basis of this framework, many scholars have proposed excellent improvement measures, which are widely used in the field of contact force-tracking. In reference [24], the authors implemented a new impedance function based on the desired force, environmental stiffness, and unknown error; directly minimized the force error through simple adaptive gains; and performed robust control algorithms on the environment and stiffness as well as on uncertainties in robot dynamic model compensation. In reference [25], the authors proposed a force-tracking impedance controller that provided a free overshoot contact force for partially unknown interactive environments. The control gain was continuously adjusted by the environmental stiffness estimated online using the extended Kalman filter to avoid force overshoot and instability. In reference [26], the author proposed to realize adaptive variable impedance control by adjusting impedance parameters online according to tracking error, and the effectiveness of this method was verified through experiments. In reference [27], the authors proposed an adaptive admittance controller based on force error information to compensate for admittance parameters in real time.

### 2.2. Deep Reinforcement Learning Method

Deep reinforcement learning combines deep learning and reinforcement learning, giving robots a way of thinking that is close to humans, enabling robots to automatically learn operational skills directly in the experience of interacting with their environment. In manipulator control research, the application based on deep reinforcement learning also shows a good control effect, such as opening a door and tightening a cap.

This is some research of deep reinforcement learning in the field of constant force-tracking. In reference [28], the authors used impedance control and reinforcement learning based on a balance-point control theory to determine the impedance parameters of contact tasks and verified the effectiveness of the method through dynamic simulations of various contact tasks. In reference [29], the authors focused on the problem that contact force was difficult to keep constant when the manipulator tracked an unknown surface workpiece, a compensation term was added to the traditional explicit force controller to find the optimal parameters through deep reinforcement learning to achieve constant force contact control. In reference [30], the authors improved the tracking performance of the nominal feedback controller using a compensation method based on reinforcement learning and achieved good results in experiments. In reference [31], the authors used reinforcement learning to study optimal online contact forces in unknown environments and used a model-free controller to guarantee position- and force-tracking. In reference [32], the author proposed an autonomous robotic ultrasonic imaging system based on reinforcement learning. The proposed force-displacement control method realizes the adaptive constant force-tracking of the ultrasonic probe against the soft target. The method has also been verified through virtual and volunteer experiments.

Compared with deep reinforcement learning methods, traditional control methods have certain limitations. Specifically, the position change of the dynamic auscultation environment is unpredictable, and the traditional control method makes it difficult to establish an accurate mathematical model, so it is difficult to achieve effective control. The method based on deep reinforcement learning does not need to establish an accurate mathematical model of the controlled robot in advance. The robot’s operating system directly controls the robot’s motion according to the information gathered from interacting with the environment, which can be easily extended to more complex applications, such as the introduction of trajectory-tracking in the process of constant force-tracking in dynamic environments. In Table 1, we summarize the related work of the above constant force-tracking.

## 3. Method

This paper proposes a constant force-tracking strategy based on ppo algorithm [19]. In the whole control process, the deep reinforcement learning method is used to complete the learning process of the entire task. Based on the learned policy network, the action is generated by observing and understanding, and the position control of the manipulator is directly controlled to achieve constant force-tracking.

### 3.1. System Overview

The overall frame diagram of constant force-tracking is shown Figure 1. In this framework, the ppo agent outputs the required position increment according to the state obtained by the current contact object and then controls the position of the manipulator to achieve a safe interactive position between the manipulator and the dynamic environment, finally achieving the purpose of controlling the contact force.

### 3.2. Environment Modeling

In order to solve the medical auscultation task with a dynamically changing position, the agent needs to perform a great deal of training in the virtual environment. Considering the diversity and time-variance of the environment, creating a more realistic simulation model is the basis of training.

#### 3.2.1. Modeling Analysis

An important aspect of environmental modeling is realizing the simulation of soft tissue deformation and force feedback. Soft tissue deformation is the physical reaction of skin under the action of external force. The geometric model of soft tissue is irregular, the deformation is complex, and the biomechanical properties are diverse. Therefore, soft tissue needs to be simplified. In elastic modeling, in order to simplify and abstract the work, five basic assumptions are used [33]: (1). it is assumed that the matter in the object is continuous; (2). it is assumed that the material in the object is uniform; (3). it is assumed that the matter in the object is isotropic; (4). it is assumed that the object is elastic; and (5). it is assumed that the displacement and deformation of the object are small. When a stethoscope is in contact with the skin, since the area of skin on the human chest and back is relatively large compared to the stethoscope, but in a small area, the curvature is relatively small, so it is simplified as an elastic body with a flat surface. Usually, the contact direction between a stethoscope and human skin is the normal direction, and the pressure of the stethoscope on the skin produces the greatest deformation on the skin when normal contact is made and the human body feels the most obvious. Therefore, the manipulator is mainly controlled in this direction, and its control mainly depends on force feedback information to show the corresponding force. In the simulation environment, when the manipulator end-effector contacts the contact surface, the contact force will cause deformation of the contact surface.

#### 3.2.2. Motion State

Another important aspect of environmental modeling is to achieve dynamic changes in location; however, everyone’s respiratory rate and physical condition are different, so the dynamic change of location is diverse. In order to meet certain authenticity in a simulation environment, the position change is realized by connecting the contact surface with sliding joints. More specifically, for the position change of the contact surface, the motion state of the sliding joint is determined via the following functions [34]: (1)$$P=\alpha sin(\frac{\pi }{\beta }).$$
where *P* denotes the moving distance of the sliding joint and α and β are constants. The former is used to determine the upper and lower limits of the motion distance of the sliding joint; in the auscultation simulation, it represents the variation range of the skin surface. The latter is used to determine the frequency of movement of the sliding joint in a motion cycle, that is, representing the respiratory rate of the patient in an auscultation simulation. According to the above requirements, Figure 2 shows the changing process of contact position.

As shown in Figure 3, in the face of different application scenarios, you can meet your own needs by modifying the stiffness value and position change frequency of the contact surface.

### 3.3. Deep Reinforcement Learning

#### 3.3.1. Background

Deep reinforcement learning is a combination of deep learning and reinforcement learning. Specifically, it combines the structure of deep learning and the idea of reinforcement learning, but its focus is more on reinforcement learning. It still solves decision problems, making reinforcement learning technology truly practical and successful in some fields.

Reinforcement learning defines any decision maker as an agent and defines everything outside the agent as an environment. The goal of an agent is to maximize cumulative rewards. It is based on the Markov decision process (MDP), in which the next state is related to the current state and the action taken. MDP is usually defined as a quadruple—state space *S*, action space *A*, reward function *R* and strategy space *P*—which is necessary for reinforcement learning. This process can be used to represent the execution process of the contact task. At each step, the agent can select an action *a* according to the current strategy π. Then, the state $s_{t}$ under the current time step *t* will change after the action is executed, enter the next state $s_{t+1}$, and reward the agent $r_{t}$ according to the state $s_{t+1}$. Among them, the policy π is the operation rule used to determine the next action, which can be deterministic or random.

The reward function is a scalar feedback signal indicating how well the action is taken. Agents receive immediate reward feedback at each time step to maximize long-term cumulative reward values rather than short-term rewards through constant trial and error in the environment. By introducing a discount factor, we can express the reward value for a time step as:(2)$$Reward_{t}=\sum_{t=0}^{T}\gamma^{t}r_{t},$$
where γ∈(0,1] is used to adjust the near-term and long-term effects, that is, how long the agent considers when making decisions.

The state value function $V_{\pi }\left(s\right)$ is used to evaluate the value of the state, which represents the expectation of the reward sum of the agent in this state until the final state. The action value function $Q_{\pi }\left(s,a\right)$ is used to evaluate the value of the action, which represents the agent’s expectation of combining the rewards until the final state after selecting the action.
(3)$$V_{\pi }\left(s\right)=E_{\pi }\left[Reward_{t}|s_{t}=s\right],$$
(4)$$Q_{\pi }\left(s,a\right)=E_{\pi }\left[Reward_{t}|s_{t}=s,a_{t}=a\right],$$

The Bellman equation described in the above equation establishes the relationship between states $s=s_{t}$ and $s^{\prime }=s_{t+1}$, and also establishes the direct relation between state value function $V_{\pi }\left(s\right)$ and action value function $Q_{\pi }\left(s,a\right)$:(5)$$V_{\pi }\left(s\right)=\sum_{a\in A}^{}\pi \left(a|s\right)Q_{\pi }\left(s,a\right),$$
(6)$$Q_{\pi }\left(s,a\right)=Reward_{s}^{a}+\gamma \sum_{a\in A}^{}P_{ss^{\prime }}^{a}V_{\pi }\left(s^{\prime }\right).$$
where $P_{ss^{\prime }}^{a}=P\left(s_{t+1}=s^{\prime }|s_{t}=s,a_{t}=a\right)$ represents the state transition probability and $Reward_{s}^{a}$ represents the reward.

We obtain the numerical solution of Bellman’s equation using the dynamic programming method as a value function, and then the agent improves the strategy by continuously optimizing the value function.

#### 3.3.2. Definition of States and Actions

The design of state space and action space can significantly improve exploration efficiency and performance of reinforcement learning algorithms. Therefore, in order to improve the learning efficiency and final learning effect, we designed the following functions.

In our study, the policy input is the state of the manipulator, including proprioceptive information of the interaction between the manipulator and the environment, mainly position information, velocity information and force information:(7)$$S=\left\{P_{z},V,F_{x},F_{y},F_{z},T_{x},T_{y},T_{z}\right\},$$

$P_{z}$ represents the position of the end effector in the *Z*-axis direction of the base coordinate system of the manipulator. *V* represents the speed of the end effector of the manipulator. *F* and *T* represent the force and torque obtained by the end effector of the manipulator in real time, respectively. The output of the strategy is the position increment of the end effector of the manipulator:(8)$$A=\left\{\Delta z\right\}.$$

The agent adjusts the movement of the manipulator in the Z-axis direction by obtaining the force information and position information between the end effector and the contact surface, so as to ensure that a constant contact force is always maintained when the contact surface changes dynamically.

#### 3.3.3. Policy Training

When using deep reinforcement learning algorithms, it is crucial to converge to the global optimal solution. In this task, we use the algorithm ppo based on the actor–critic framework, which has shown good performance in many reinforcement learning continuous control tasks. Compared with the previous trust region policy optimization (TRPO) [35] algorithm, it is easier to realize.

At the same time, in order to improve the sample efficiency of the on-policy method, the importance sampling method is adopted to solve the problem that the parameters can only be updated once for each sampled datum. However, it can be seen from importance sampling that if you want to update multiple policies with the same data, then multiple policies must be in the same policy distribution interval. Therefore, the truncation coefficient ε is used to limit the gap between the new strategy and the old strategy, that is, the gradient update of the parameters is controlled. Then the objective function of ppo [19] is:(9)$$L_{ppo}=min\left[\frac{\pi_{\theta }\left(a|s\right)}{\pi_{\theta_{k}}\left(a|s\right)}A^{\pi_{\theta_{k}}}\left(s,a\right),clip\left(\frac{\pi_{\theta }\left(a|s\right)}{\pi_{\theta_{k}}\left(a|s\right)},1-\varepsilon ,1+\varepsilon \right)A^{\pi_{\theta_{k}}}\left(s,a\right)\right],$$

Here, the parameters of importance sampling are mainly truncated, where $\pi_{\theta }$ and $\pi_{\theta_{k}}$ represent the current and previous strategies, respectively, and $A^{\pi_{\theta_{k}}}\left(s,a\right)$ is the advantage function, which is optimized using the generalized advantage estimator (GAE) [36].

When the advantage is positive, the importance sampling term limits the maximum value to 1+ε, that is, the parameter controls the maximum step size when updating the gradient, and the objective function is:(10)$$L_{ppo}=min\left(\frac{\pi_{\theta }\left(a|s\right)}{\pi_{\theta_{k}}\left(a|s\right)},\left(1+\varepsilon \right)\right)A^{\pi_{\theta_{k}}}\left(s,a\right),$$

Similarly, when the advantage is negative, the importance sampling term limits the minimum value to 1−ε, that is, the parameter controls the minimum step size when the gradient is updated, and the objective function becomes:(11)$$L_{ppo}=min\left(\frac{\pi_{\theta }\left(a|s\right)}{\pi_{\theta_{k}}\left(a|s\right)},\left(1-\varepsilon \right)\right)A^{\pi_{\theta_{k}}}\left(s,a\right).$$

#### 3.3.4. Reward Shaping

The observations of reinforcement learning mainly include the force information between the end of the manipulator and the contact surface, and the factors that affect the interactive force information also include the relative positional relationship between the manipulator and the contact surface. Therefore, the design of the reward function is very important. It greatly affects the efficiency of reinforcement learning training and the effect of final force-tracking. Here we use the success reward function, the force error reward function, and the range-limited reward function to form the reward function:(12)$$Reward=R_{success}+R_{force}+R_{restrict},$$

Among them, the force error function consists of two parts. The first part calculates the error between the real-time contact force and the desired contact force. Under the excitation of this function, the manipulator constantly adjusts its action to make the contact force close to the desired force. The closer the real-time contact force is to the desired contact force, the greater the reward it obtains. The second part calculates the error between the real-time contact force and the previous contact force. In the contact process, the contact force will inevitably fluctuate due to the change of position. So under the excitation of this function, the contact force fluctuations will continue to decrease, which can be seen in Equation (Equation 13):(13)$$R_{force}=\frac{1}{|F_{now}-F_{desire}|}-0.1*\left|F_{now}-F_{last}\right|,$$

At the same time, in order to avoid the occurrence of a great reward value due to the extremely small error when the real-time contact force is constantly approaching the desired force, in the success reward function, the error value of the expected force is limited, and the success reward can be obtained by reaching the error range of the expected force. The success reward value is set relatively high in the hope of promoting the manipulator to better complete the force-tracking task during the learning process, which is the most important goal. This can be seen in Equation (Equation 14):(14)$$R_{success}=\mu ,$$

The range limit of the reward function is to ensure the efficiency of training and limit the range of agent exploration. Once it exceeds the set range, the task will end and start the next time step of learning, which will avoid some useless and incorrect situations. This can be seen in Equation (Equation 15):(15)$$R_{restrict}=-\nu .$$

The pseudo-code of the constant force-tracking control algorithm (Algorithm 1) is as follows:
**Algorithm 1** Constant Force-Tracking Control      **Input:**state: $P_{z}$,*V*,$F_{x}$,$F_{y}$,$F_{z}$,$T_{x}$,$T_{y}$,$T_{z}$      **Output:**action: Δz1:**Initialization:** policy parameters $\theta_{0}$, value function paramaters $\phi_{0}$2:**repeat**3:      Run policy $\pi_{\theta_{t}}$ in the environment for *t* timesteps4:      Take action $a_{t}$, get reward $R_{t}$ and next state $s_{t+1}$5:      Computer advantage estimates ${\hat{A}}_{t}$ based on the current value function $V_{\phi_{t}}$6:      Update $\theta_{t+1}$7:      Update $\phi_{t+1}$8:**until**$|f_{e}-f_{d}|<\delta$ (δ is the set force deviation value )

## 4. Simulation

This section focuses on the simulation evaluation of the constant force-tracking algorithm based on deep reinforcement learning. Firstly, the settings of the simulation environment are introduced. Then, the effectiveness of the proposed algorithm is verified through simulation experiments. Finally, the trained model is tested using comparative experiments and generalization experiments.

### 4.1. Simulation Process

#### 4.1.1. UR5E Robot

The UR5E is a highly flexible, 6-degrees-of-freedom (DoF) manipulator produced by Universal Robots. It has a bearing capacity of 5 kg, a radius of 850 mm, and a weight of 20.6 kg. It is significantly lighter compared to industrial robots with similar workspaces and payloads. The robot has an internal embedded controller to compensate for various nonlinearities such as gravity, coriolis effects, etc., allowing repetitive, risky tasks to be converted into safe, automated operations.

#### 4.1.2. Environment Setting

Firstly, the force-tracking control environment under dynamic positioning is established using the simulation environment of pybullet. The environment consists of a UR5E manipulator, force sensor, and tracking target. The tracking target is a manipulator connected by translational joints, which is used to simulate the state of dynamic fluctuation of position. As shown in Figure 4, we use the 6-DoF cooperative robot to perform this task, which is controlled via python language. At the same time, the six-dimensional force sensor can obtain the force and torque in the contact process in real time. Finally, we verify the effectiveness of the algorithm by analyzing the force error between the expected force and the contact force at the end of the manipulator.

In the simulation experiment, we need to set some parameters. The tracking target of the manipulator is a plane with dynamic position changes. Its stiffness is set to 3800 N/m, the position change range is ±1 cm, and the position change frequency is set to be sampled 2000 times by the sensor at the end of the manipulator in each cycle. The expected force of tracking is set to 5 N, which can be changed according to the specific task requirements. The two parameters *u* and *v* in the reward function are set to 20 and −100, respectively.

### 4.2. Experimental Results

#### 4.2.1. Proposed Constant Force-Tracking System

We trained 4100 sets in the simulation environment, and the training process took about 8 h. The training results are shown in the figure below, including the reward function, contact force information, and contact surface deformation. The experimental results show that the manipulator performs well in the whole process of constant force-tracking after training.

As shown in Figure 5, rewards increase very quickly in the early stages of training, but in the later stages, reward values increase at a slower pace. This shows that the manipulator can quickly adjust its own state and control the contact force to reach the vicinity of the expected force. However, due to the constant change of the environmental position, a small error in the relative position between the manipulator and the contact surface will lead to a fluctuation of the contact force, so it cannot be completely rewarded for success, and the later growth is slow. It can be seen from Figure 6 that the obtained contact force fluctuates within a small range of the expected force, for which the amplitude is small. At the same time, under the action of contact force, the contact surface produces about a 0.13 cm elastic deformation, as shown in Figure 7.

#### 4.2.2. Methods Comparison

In order to further illustrate the feasibility of our proposed method to achieve constant force-tracking in a dynamic environment, we compare it with two methods in the field of constant force control, namely the force-control algorithm based on deep reinforcement learning and the adaptive variable impedance control algorithm.

The force-control algorithm is based on deep reinforcement learning. In order to ensure that the normal force at the end of the robot is constant, a compensation term is added to the traditional display force controller, and the optimal parameters are found via a deep reinforcement learning method to meet the robot’s tracking scene. The specific control formula [29] is as follows:(16)$$\begin{array}{cc} e_{k} & =f_{e}-f_{d},\\ \Delta_{k} & =k_{p}\times e_{k}+k_{d}*{\dot{e}}_{k}+\delta_{k}sgn\left({\dot{e}}_{k}\right)+\varsigma_{k}sgn\left(e_{k}\right),\\ \Delta y_{k} & =\Delta y_{k-1}+\Delta_{k}. \end{array}$$
where $\delta_{k}$ and $\varsigma_{k}$ are compensation parameters, which are learned through deep reinforcement learning.

The adaptive impedance control algorithm, whose impedance parameters change according to the force feedback, realizes constant force-tracking in uncertain environments. The specific control formula [26] is as follows:(17)$$f_{e}\left(t\right)-f_{d}\left(t\right)=m\ddot{e}\left(t\right)+\left(b+\Delta b\left(t\right)\right)\dot{e}\left(t\right),$$
(18)$$\left\{\begin{array}{ccc} \Delta b(t) & =\frac{b}{\dot{e}\left(t\right)}\Phi \left(t\right), & \\ \Phi (t) & =\Phi \left(t-\lambda \right)+\sigma \frac{f_{d}\left(t-\lambda \right)-f_{e}\left(t-\lambda \right)}{b}. \end{array}\right.$$
where λ is the sampling period of the controller and σ is the update rate.

Under the same sampling frequency and the same stiffness condition, the strategy after training is tested separately, and the final contact force is shown Figure 8. The experimental results show that these three methods can complete the constant force-tracking task effectively. Compared with the proposed method, it is obvious that method [26] and method [29] produce greater contact force fluctuations.

#### 4.2.3. Generalization Experimental Results

In this section, we change the scenario configuration to test the performance of the model. We consider that, in a medical auscultation setting, the chest goes up and down once every time the normal adult breathes, such time being approximately 3–5 s, including the time to inhale and exhale. The sampling frequency of the UR5E manipulator is 500 HZ, so the contact force is updated between 1500 and 2500 times per breath. Therefore, in the simulation environment, we set the change frequency of the contact surface in each motion cycle to 1500 times and 2500 times, respectively. Secondly, because the skin state of each patient is also different, we change the stiffness of the contact surface to 1500 N/M and 5000 N/M in the simulation environment. According to the above information, the effect of the model test is shown in Figure 9.

It can be seen from the figure that this strategy shows good, constant force-tracking performance. Facing an unknown dynamic environment, the stiffness of the environment and the change frequency of the target plane position are negatively correlated with the effect of constant force-tracking. Rapidly changed position or large environmental stiffness leads to a significant contact force. However, the contact force is always kept within the range of ±0.5 N of the expected force, showing constant force-tracking performance under different conditions.

## 5. Real Experiment

In order to further confirm the practicability of the algorithm, we also built an experimental environment in a real setting. Constant force-tracking performance in this dynamic environment will also be verified in the actual auscultation operation. As shown in Figure 10, the auscultation operation is performed using a 6-degree-of-freedom UR5E manipulator. The working contact surface and the contact area contact each other, and the manipulator is adjusted by the contact force obtained in real time.

### 5.1. ROS-Based Control Framework

Based on the Ubuntu system and ROS, we apply the trained model to the control of the cooperative robot and test the model. Its overall architecture is shown in the Figure 11:

### 5.2. Experienment Results

The control strategy after training is tested. In the constant force-tracking experiment of the dummy, the manipulator first moves vertically downward. After obtaining the contact force, the constant force-tracking control is started. The specific experimental results are shown in Figure 12. Compared with the simulation results, the force error is relatively enlarged, but in general, better constant force-tracking performance can eventually be achieved.

### 5.3. Discussion

As a decision-oriented control method, deep reinforcement learning directly establishes the relationship between output and tasks. Therefore, it has potential application prospects in complex medical scenarios. In this paper, we propose a constant force-tracking control method based on deep reinforcement learning, which is mainly used in medical auscultation scenarios. In a virtual environment, we trained the control strategy for the dynamic auscultation setting and verified the control strategy through simulation and real experiments. The final effect is in line with our initial expectations and can better achieve constant force-tracking. Of course, in a real clinical environment, the stethoscope may need to contact the surface in a specific posture, so one of our next studies will expand the dimension of the action space to achieve more complex medical environment applications.

## 6. Conclusions

In this paper, we propose a manipulator force-control strategy based on a ppo algorithm for medical auscultation tasks in dynamic environment. At the same time, in order to better realize the training of the manipulator, we design a modeling method to meet the physical characteristics of the dynamic setting. In this simulation environment, considering that each person’s respiratory rate and physical condition are different, the real environment can be simulated by changing the stiffness and position change frequency of the contact surface, and a large number of simulation tests are carried out on the trained strategy. From the experimental results, it can be seen that the manipulator can complete the constant force-tracking task effectively in the dynamic auscultation setting.

In future work, we will focus on real-world applications. Then, we will explore controlling the manipulator to accomplish other more complex tasks, such as trajectory tracking on a plane with dynamically changing positions.

## Figures and Tables

**Figure 1 sensors-23-02186-f001:**
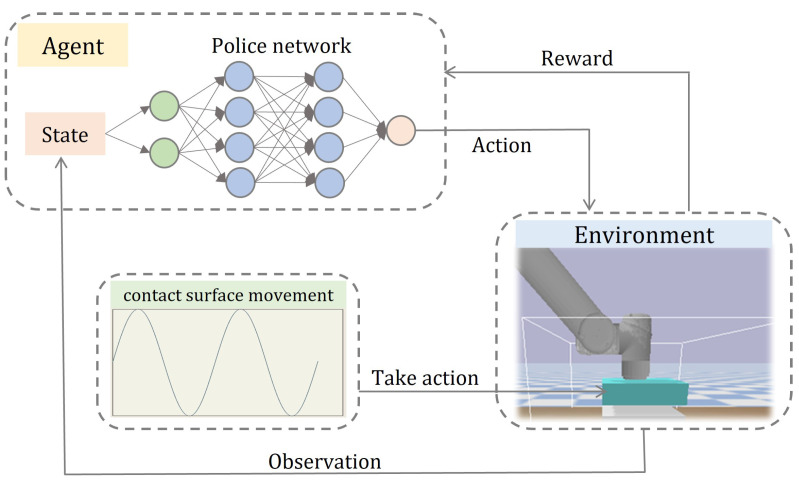
Framework of auscultation strategy. It consists of two parts: a deep reinforcement learning module and a dynamic location module. First, the dynamic position module realizes the contact surface position change. Then, the deep reinforcement learning module outputs the manipulator action based on the acquired manipulator state and obtains a reward according to the newly acquired state.

**Figure 2 sensors-23-02186-f002:**
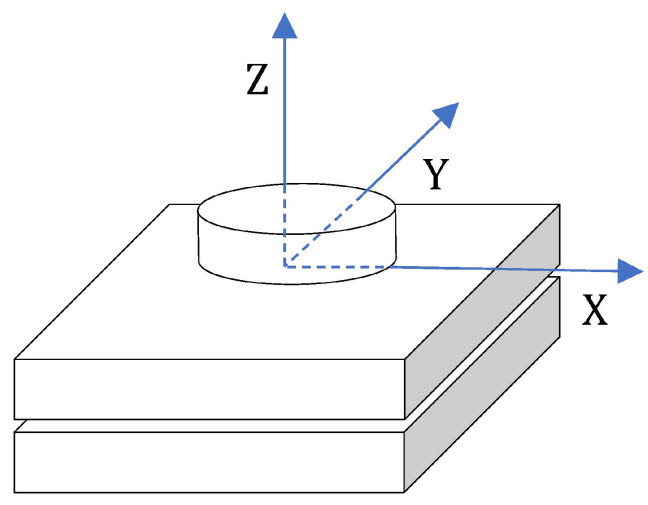
Contact stethoscope and simulation surface.

**Figure 3 sensors-23-02186-f003:**
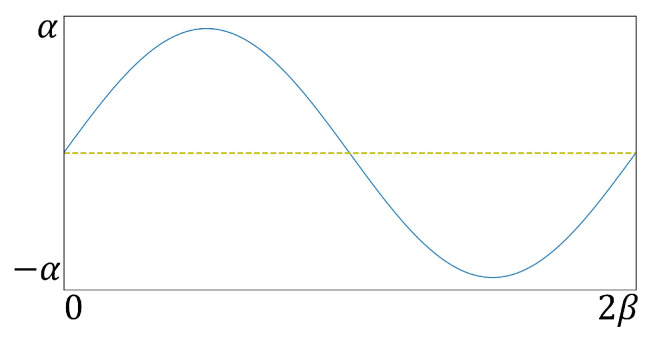
Contact surface motion curve.

**Figure 4 sensors-23-02186-f004:**
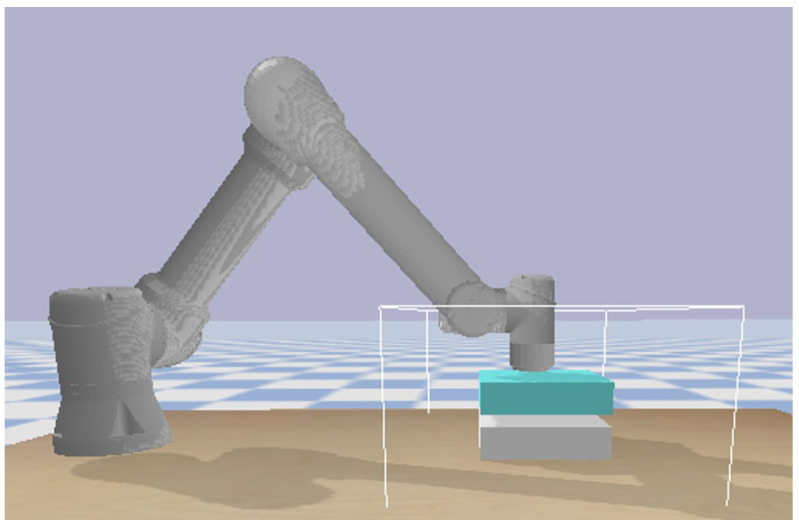
Simulation environment.

**Figure 5 sensors-23-02186-f005:**
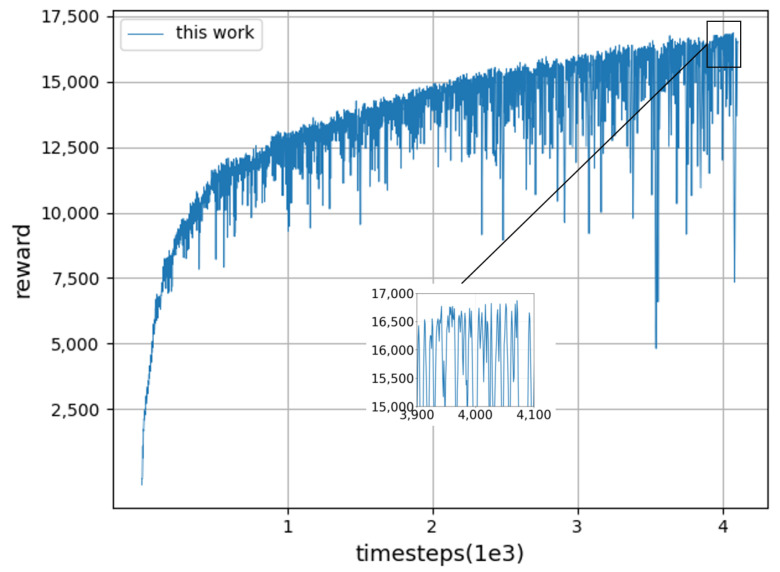
Training curve of constant force-tracking task.

**Figure 6 sensors-23-02186-f006:**
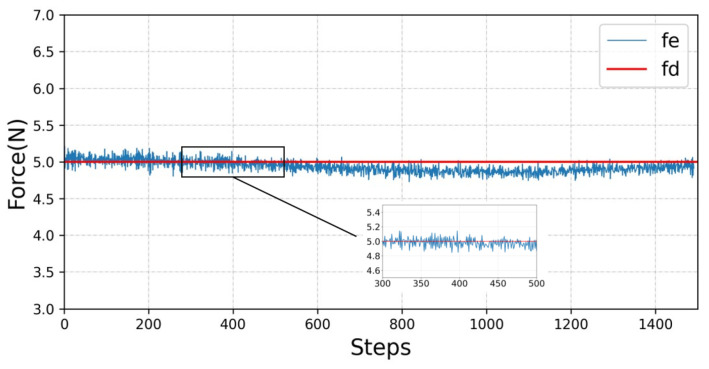
Constant force-tracking performance. Where $f_{e}$ represents the desired contact force and $f_{d}$ represents the acquired real-time contact force.

**Figure 7 sensors-23-02186-f007:**
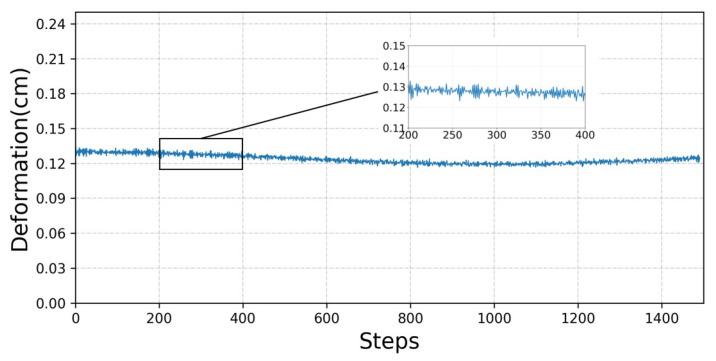
Deformation size of contact surface.

**Figure 8 sensors-23-02186-f008:**
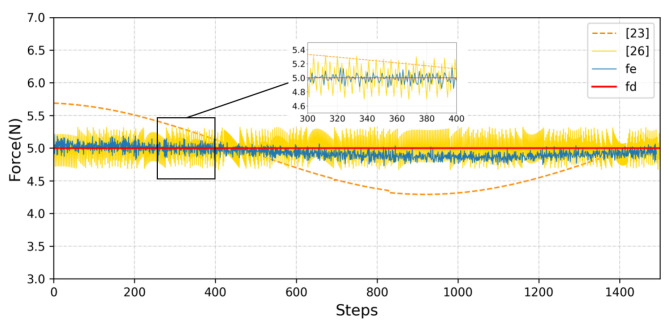
Performance comparison where [23] represents the contact force obtained using the manipulator under the control of the adaptive variable impedance method and [26] represents the contact force obtained using the manipulator under force control based on deep reinforcement learning.

**Figure 9 sensors-23-02186-f009:**
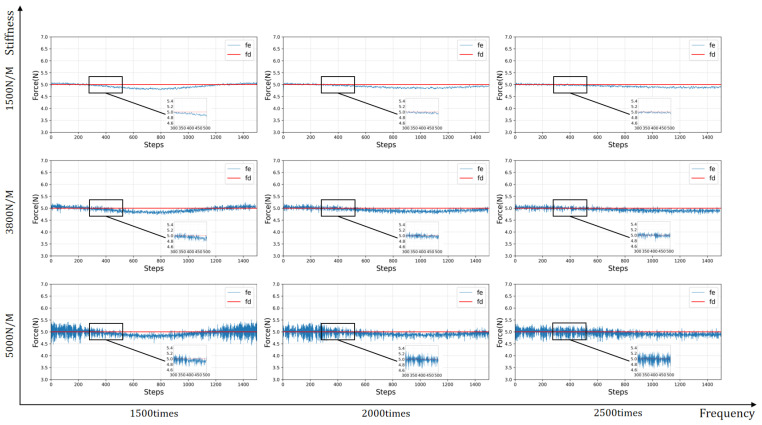
Performance testing. By changing the stiffness and frequency of the contact surface, the trained model is tested. The results show that the model has good generalization.

**Figure 10 sensors-23-02186-f010:**
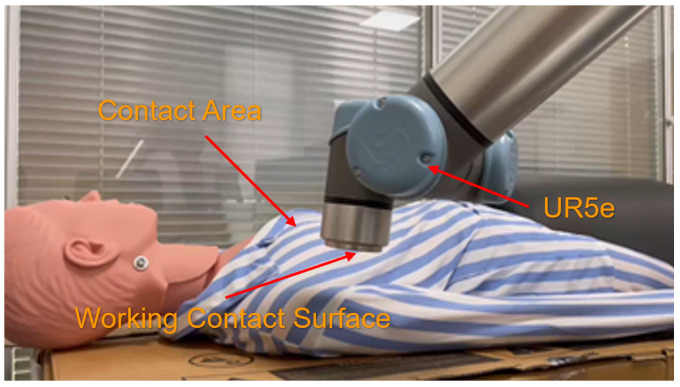
Construction of auscultation scene.

**Figure 11 sensors-23-02186-f011:**
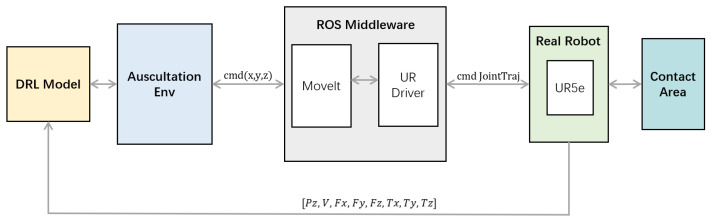
Control framework.

**Figure 12 sensors-23-02186-f012:**
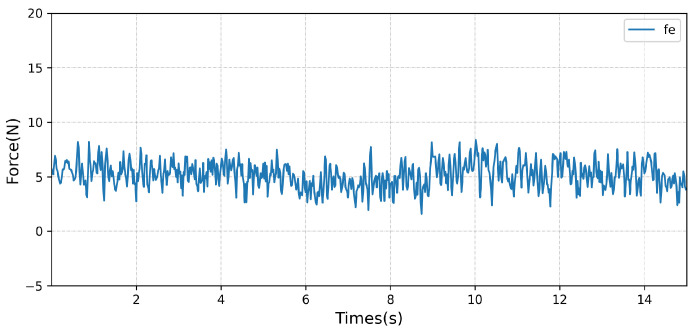
Real experimental results.

**Table 1 sensors-23-02186-t001:** Summary and comparison of impedance control methods and deep reinforcement learning methods.

Reference	Area	Methodology	Main Remarks
[24]	impedance control	Robust position control algorithm is used to compensate for the uncertainty in robot dynamics	Good response to environmental stiffness mutation to achieve stable force-tracking
[25]	impedance control	Free overshoot for partially unknown interactive environments	The control logic can handle both space proximity motion and contact tasks, avoiding force overshoot and instability
[26]	impedance control	Adaptive variable impedance control based on tracking error online adjustment of impedance parameters	It has better force-tracking performance than constant impedance control
[27]	impedance control	Adaptive admittance control based on force error information timing compensation admittance parameters	It provides an effective solution for force control of basic operations in uncertain environments
[28]	deep reinforcement learning	Impedance control based on equilibrium point control theory and reinforcement learning determines the parameters of impedance control	The performance of the contact task is optimized in uncertain environments
[30]	deep reinforcement learning	Uses additive compensation to correct the control input given by a nominal controller	The RL compensation method significantly reduces the tracking error
[31]	deep reinforcement learning	Reinforcement learning is used to learn the optimal contact force online	Effectively completes the robot interactive control in unknown environments
[32]	deep reinforcement learning	Set the contact force as the action space of the end-effector and map the contact force to the lowest-level robot command	Achieve adaptive constant force-tracking of soft targets
This work	deep reinforcement learning	A dynamic auscultation environment is designed, and DRL is used to learn constant contact force online	In response to changes in environmental position, stable force-tracking is achieved

## Data Availability

Not applicable.

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
