# Peer review of "Constant Force-Tracking Control Based on Deep Reinforcement Learning in Dynamic Auscultation Environment"

_sensors, 2023, doi:10.3390/s23042186_

Round 1

Reviewer 1 Report

The paper presents two major pitfalls:

1. Introduction and abstract need to be more convincing. 

1a. For the abstract, I suggest to report main novelties and quantitative results.

1b. For the introduction I suggest the authors to include explicit paragraphs about i) motivations, ii) objectives, iii) deficiency in the literature, iv) gaps filled in the literature, v) main novelties and contributions to the research field, vi) the approach in brief and the structure of the paper clickable to reach specific sections.

2. The paper really lack a literature review/related work section. The authors must add a robust related work section where they compare against papers in the literature point-to-point. A table summarizing primary remarks of each paper surveyed in the section will close the section providing a way to grasp all the needed information at a glance. Images are welcome. From this section, the reader must understand what has been done in the literature and what is your novelty.

Other important aspects are:

3. Authors should explain why they used Deep reinforcement learning over other techniques like https://doi.org/10.1016/j.eswa.2022.118302

4. The paper needs further sections: i) discussion, ii) limitations and future works, iii) expected impact, iv) application scenarios

5. The authors should compare their performance with those obtained by other paper if possible directly, otherwise indirectly.

6. I wondering which kind of validation approach the authors employed in their experimental part. Please, add details about it (see https://doi.org/10.1007/s00521-022-07454-4)

7. Please add details on the dataset used and the pseudocode (in the appendix perhaps)

Reviewer 2 Report

In this paper, a constant force tracking control method for medical listening robots based on deep reinforcement learning is proposed, a model satisfying the physical characteristics of dynamic scenes is established to simulate the dynamically changing environment with breathing, and a task-based optimal reward function is designed to achieve efficient learning of robot control strategies, and finally the error between the actual tracking force and the desired force of the proposed method is verified by simulation experiments to be within 0.5N or less. To improve the paper, I have the following suggestions:

(1)In the experimental part of this paper, only simple simulations were performed. In order to better reflect the feasibility of the proposed method, it is hoped that specific experimental contents can be added.

(2)In this paper, an up-and-down moving plane is used instead of the human surface to simulate the human breathing process, but the deformation characteristics of the human skin under the force state are not mentioned.

(3)In this paper, equation (1) is used to represent the skin heaving during breathing, However, the change in air pressure in the lungs during breathing cannot be expressed by a simple sine function.

(4)What do the parameters to the right of the equal sign of equation (14) and equation (15) specifically represent?

(5)Lines 167 and 168 in the paper show that the next state is only related to the current state and not to the previous state. Does this lead to a local optimal solution as a result of the proposed algorithm?

Round 2

Reviewer 1 Report

The authors have responded in a complete manner, the work can be published.

Author Response

Thanks.